# Hot Deformation Behavior of TA1 Prepared by Electron Beam Cold Hearth Melting with a Single Pass

**DOI:** 10.3390/ma16010369

**Published:** 2022-12-30

**Authors:** Zhibo Zhang, Weiwei Huang, Weidong Zhao, Xiaoyuan Sun, Haohang Ji, Shubiao Yin, Jin Chen, Lei Gao

**Affiliations:** 1Faculty of Metallurgical and Energy Engineering, Kunming University of Science and Technology, Kunming 650093, China; 2Science and Technology Innovation Department of Kunming Iron & Steel Co., Ltd., Kunming 650302, China

**Keywords:** electron beam cold hearth melting, hot deformation, TA1, constitutive model, processing maps

## Abstract

The Gleeble-3800 thermal simulator was used for hot compression simulation to understand the hot deformation performance of TA1 prepared by the single-pass electron beam cold hearth (EB) process. The deformation degree is 50% on a thermal simulator when the temperature range is 700–900 °C, with a strain rate of 0.01–10^−1^ s. According to the thermal deformation data, the true stress-strain curve of TA1 was studied. Meanwhile, the constitutive model and processing map were established through the experimental data. These results indicate that the deformation temperature negatively affects strain rate and flow stress. The heat deformation activation energy of EB produced TA1 sample was lower than that of VAR produced TA1 sample in the studied range. The best processing areas of EB-produced TA1 were strain rates of 0.05–0.01 s^−1^, within 700–770 °C; or strain rates of 0.01–0.15 s^−1^; 840–900 °C. The results of this paper enrich the fundamental knowledge of the thermal deformation behavior of TA1 prepared by EB furnaces.

## 1. Introduction

Titanium is an important metal applied in modern industry and has characteristics including high strength, lightweight, strong corrosion resistance, and good temperature endurance [1,2]. The TA1 plate is generally obtained by rolling titanium slab as the most used commercial titanium material. However, the shape and surface quality control of the TA1 titanium slab is problematic, resulting in the high cost of the TA1 plate [3]. To reduce the production cost, the electron beams cold hearth melting technique (EB) was introduced to produce TA1 slab, with which high-quality TA1 products can be formed with single pass melting and casting. Compared with the traditional vacuum consumable electrode arc (VAR) furnace melting technology which requires several passes, the titanium slab melted by a single pass EB furnace is more energy saving. The prepared EB product can be used for direct rolling without forging and perforation molding [4,5]. Meanwhile, the EB technique was known to have a strong purification ability, especially for high-density inclusion (HDI), which can hardly be removed by the VAR technique [6,7]. Thus, the EB technique can reduce titanium waste, reducing the production cost of titanium plates [8].

However, in the actual production process, TA1 titanium slabs using EB furnaces usually adopt the same hot rolling process scheme as the TA1 titanium slab melted by the VAR furnace. Since the thermal deformation behaviors of TA1 slabs produced using EB and VAR are different, surface defects, including peeling and scratching, appear during the hot rolling process of TA1 slabs melted by EB furnace, which increases the edge cutting process of titanium coil and production cost.

Many scholars have studied the thermal deformation behavior of TA1 slabs melted by a VAR furnace. Ma et al. [9] established a processing map for the TA1 plate melted by VAR. The results showed that TA1 slab was suitable for rolling under the conditions of a strain rate at 5 s^−1^, a temperature at 700–750 °C, and the deformation of each pass is greater than or equal to 25%. Li et al. [10] conducted thermal compression simulation experiments on the intermediate billet of hot-rolled TA1 titanium melted by VAR, and the processing map was formed. The best conditions for smelting TA1 in the VAR furnace included a strain rate and temperature range of 1–20 s^−1^ and 750–850 °C. However, few studies are devoted to the thermal deformation behavior of TA1 slabs melted by the EB furnace. Huang et al. [11] researched the thermal deformation performance of TA1 slabs prepared by residual titanium in an EB furnace. The influences of thermal deformation on the microstructure were discussed, whilst the impacts of thermal deformation on rheological stress were not considered. Thus, it is necessary to study the influence of rheological stress during the hot deformation of TA1 to find out the hot deformation characteristics of EB furnace casting billet and effectively predict and formulate the thermoplastic processing technology of TA1.

Therefore, this manuscript researches the thermal deformation behavior of TA1 prepared by EB furnaces, focusing on the impacts of strain rate and deformation temperature on the flow stress. Meanwhile, the constitutive model and processing map are founded according to the stress-strain data, which provides a reference for the improvement of the EB furnace melting technique and the subsequent TA1 hot rolling process.

## 2. Materials and Methods

The test material was an industrial TA1 slab produced by an EB furnace in a factory in Kunming, China. The chemical composition is indicated in the quality assurance certificate of the enterprise, and has been verified according to the GB/T4698 standard method [12]. The chemical composition is displayed in Table 1 [13]. The Gleeble sample, sized of Φ 8 × 12 mm, is cut from a single pass EB melted slab using wire cutting. To ensure the uniformity of the samples, 10cm of the slab was removed from the end before cutting the sample. After sampling, Gleeble-3800 was used for the thermal simulation compression test samples. The initial microstructure of the material before the Gleeble test is shown in Figure 1. It can be seen from the figure that the original structure of the single pass EB furnace direct melting billet is equiaxed α phase. Still, the grain structure is coarse, and the average grain size is about 3–5 cm, reaching the centimeter level.

The heat treatment temperatures of the sample were selected at 700 °C, 750 °C, 800 °C, 850 °C, and 900 °C, according to the thermal simulation compression deformation temperature, combined with TA1 phase transformation temperature and empirical rolling temperature range. The technical details about mechanical testing are: The temperature is measured by a thermocouple fixed in the middle of the sample. The temperature was recorded by the data acquisition system of the thermal simulator. The heating rate is 20 °C/s. Since the technology of hot compression deformation test using Gleeble testing machine is relatively mature, using one sample for each specific temperature and strain rate can meet the research requirements. But it also depends on the ability of the tester to control the machine. When the sample deformation is not uniform, that is, the obtained sample has irregular “drum shape” or even non- “drum shape”, the collected data will also appear obvious anomalies. At this point, another sample will be made to correct the corresponding process.

The corresponding strain rates were 0.01 s^−1^, 0.1 s^−1^, 1.0 s^−1^, 5.0 s^−1^, and 10 s^−1^, separately. 50% was the set value of compression deformation. The settled temperature is maintained for 5 min, and water-cooling was carried out immediately after compression treatment.

## 3. Results

### 3.1. True Stress-Strain Curve of TA1 Sample Produced by Single Pass EB

The friction effect in the process of thermal deformation is reduced by placing graphite sheets at both ends of the sample, the most important variables are the friction force and the geometry of the sample [14,15]. To reduce the influence of friction, a piece of 0.5 mm tantalum foil was placed at the ends of the cylindrical sample during the compression test. The flow stress is the stress that must be applied to cause a material to deform at a constant strain rate in its plastic range. Individual data with obvious deviation were removed when processing the flow stress curve. Figure 2 displays the true stress-strain curve of the TA1 sample at various deformation temperatures and strain rates. With the rise of strain rate, the flow stress increases at a certain deformation temperature from Figure 2. For example, at 0.01 s^−1^ strain rate and 700 °C, the maximum flow stress is about 58 MPa. While the maximum flow stress is about 160 MPa at a 10 s^−1^ strain rate. At high strain rates, the strain hardening trend is obvious, and the degree of dynamic softening is low, so the flow stress is enormous. Figure 3 shows the typical metallographic structure at 750–900 °C and 1 s^−1^ strain rate. When deformed below 850 °C, α-phase equiaxed polyhedral grain structure is obtained, which is mainly different in grain size and uniformity. The non-uniformity of grains obtained after deformation at 750 °C is obvious. It can be seen that some grains have not grown up after recrystallization. Then, with the increase of temperature, the grains grow up gradually and distribute uniformly in the region. However, when the temperature reaches 900 °C for deformation, as the deformation temperature is already in the β-phase zone, the sample structure has a flaky nature, the grains are irregular, and the sawtooth structure of the β-phase grain boundary appears. So, when the strain rate is low, the flow stress is low, which is because the sample is affected by dynamic recovery and dynamic recrystallization softening [16]. Under the condition of constant strain rate, the flow stress declines with the rise of deformation temperature. For example, when the temperature is 700 °C, the maximum flow stress is about 160 MPa at the strain rate of 10 s^−1^, while 30 MPa is the maximum flow stress at 900 °C. Therefore, when the deformation temperature grows gradually, the softening impact of TA1 is more obvious, and the trend of flow stress reduction is more significant.

At high-temperature compression deformation, the relationship between peak stress and strain rate as a function of temperature is presented in Figure 4a,b. With a constant temperature, the peak stress grows with the rise of the strain rate as shown in Figure 4a; With a constant strain rate, the peak stress reduces with the increase of temperature, as shown in Figure 4b. When the temperature is 700–850 °C, the sensitivity of peak stress to strain rate tends to be consistent. However, when the temperature reaches 900 °C, the sensitivity of peak stress decreases obviously, and the strain rate changes tend to be gentle. This can be attributed to the fact that with the increase in temperature, the average atomic kinetic energy of the material increases, and the thermal activation increases, which reduces the critical shear stress of grain boundary slip and weakens the resistance to slip and dislocation [17].

During the hot deformation of TA1, the hardening impact of strain hardening, dynamic recovery, and dynamic recrystallization softening mechanism will cause the variation of flow stress. The stress-strain curve has two stages: unsteady deformation and steady deformation. At the initial stage of deformation, namely unsteady deformation, the flow stress growths quickly within a very small range of strain increases; The strain endures to growth while the flow stress grows gradually in the steady-state deformation process. The flow stress rises slowly when the deformation temperature is at a low level, and the strain rate is high. The flow stress grows slowly to the peak value, then declines slowly, and finally tends to be stable under the condition of a low strain rate and high deformation temperature. Plastic deformation occurs in the stage of unsteady deformation. The interaction of epistatic faults in the slip system causes the dislocation density to increase speedily, resulting in dislocation entanglement and stacking, and finally forms cellular dislocation structure, resulting in higher stress for material deformation [18]. Hardening is dominant, and the flow stress grows rapidly in the stage of unsteady deformation.

As the strain increases, the material accumulates more energy in the steady-state deformation stage. Cellular dislocations form substructures, and dislocations of opposite signs are canceled out. The softening influence of dynamic recovery rises continuously, and the growing trend of flow stress becomes gentle. The material will have enough time to accumulate more energy at a large deformation temperature and a small strain rate [19]. Dynamic recrystallization will arise during the deformation after the material accumulates enough energy. At this time, the softening effect is more pronounced. The flow stress declines slowly after getting the peak value. Finally, the hardening effect affected by the rise of strain is balanced with the softening effect of dynamic recovery and dynamic recrystallization, and the flow stress stays unwavering [20]. However, with a low deformation temperature and a significant strain rate, there exists not enough time to accumulate energy for the material. Simultaneously, the temperature cannot reach the critical temperature required for dynamic recrystallization. Therefore, the softening effect of the material is almost not noticeable, which shows that the flow stress continues to increase slowly.

### 3.2. Constitutive Model of TA1 Sample Produced by Single Pass EB

When studying the mechanical properties of metal materials during plastic deformation, the constitutive model can frequently describe the relationship between strain, stress, temperature, and strain rate. As stated by stress-strain data of TA1 melted and cast in the EB furnace obtained in the experiment, the constitutive model was derived by using the constitutive equation. Arrhenius constitutive equation was exploited to derive, which was summarized and improved from various models, and can accurately express the relationship between flow stress, strain rate, and temperature [21]. The three expressions of the Arrhenius equation are present in Equations (1)–(3). Equation (1) applies to low stress where the strain rate and stress are power functions; Equation (2) applies to the high-stress situation where there is an exponential connection between strain rate and stress; When the connection of strain rate and stress remains a hyperbolic sine function, Equation (3) is applicable.
(1)$$\dot{\varepsilon }{=A}_{1}\sigma^{n_{1}}exp\left(-\frac{Q}{RT}\right)$$
(2)$${\dot{\varepsilon }=A}_{2}\exp\left(\beta \sigma \right)\exp\left(-\frac{Q}{\mathrm{RT}}\right)$$
(3)$$\dot{\varepsilon }=A\left[sinh{\left(\alpha \sigma \right)}^{n}\right]exp\left(-\frac{Q}{RT}\right)$$
where, $\dot{\varepsilon }$ represents the strain rate, (s^−1^); σ represents the stress, (MPa); T stands for deformation temperature, (K); *Q* means the activation energy of thermal deformation, (J/mol); *R* means the gas constant, (J/mol·K), the value indicates 8.314 [22]; *A*, *A*_1_, *A*_2_, n, n_1_, β, α represent the material constant, and α
*=*
$\frac{\beta }{n_{1}}$.

The natural logarithms were taken on both sides of Equations (1)–(3), and the equations were transformed into Equations (4)–(6). The lnσ was settled as abscissa, $\ln\dot{\varepsilon }$ was settled as the ordinate, the graph was drawn in Figure 5a, where the true strain was 0.69; With σ was settled as abscissa, and $ln\;\dot{\varepsilon }$ was settled as the ordinate, the graph with the true strain of 0.69 was drawn in Figure 5b. In Figure 5a, n_1_ represented the average slope of the five straight lines (σ-$\ln\;\dot{\varepsilon }$), and the average slope of the five straight lines in Figure 5b is *β*. n_1_ was reckoned to be 4.89534, *β* was 0.106564, and the *α* was 0.021768.
(4)$$ln\;\dot{\varepsilon }{=n}_{1}ln\sigma +lnA_{1}-\frac{Q}{RT}$$
(5)$$ln\;\dot{\varepsilon }=\beta \sigma {+lnA}_{2}-\frac{Q}{RT}$$
(6)$$ln\;\dot{\varepsilon }=nln[sinh(\alpha \sigma )]+ln\;A-\frac{Q}{RT}$$

The relationship of $ln\left[sin\;h\left(\alpha \sigma \right)\right]-ln\;\dot{\varepsilon }$ is drawn into a curve in Figure 6a. The average n of the straight-line slope of ln[sinh(ασ)] was calculated, which was 2.982678. In a certain range, the activation energy of thermal deformation is determined when the strain rate $\dot{\varepsilon }$ is certain. The partial derivative of 1/*T* on both sides of Equation (6) was calculated, expressed as [23]: (7)$$Q=Rn\frac{\partial ln[sinh(\alpha \sigma )]}{\partial \left(\frac{1}{T}\right)}$$

Figure 6b shows the relation curve of ln[sin h(ασ)]−1000/*T* when the true strain was 0.69. Under different strain rates, the average straight-line slope value was 16.55988, which was substituted into Equation (7) to reckon the thermal deformation activation energy (*Q*), which was 410.655 kJ/mol.

In general, when characterizing the impact of temperature and strain rate on the deformation behavior of materials, the Zener-Holloman parameter of temperature-compensated strain rate factor is often used in Equation (8) [24].
(8)ln Z=ln A+n ln[sinh(ασ)]

According to Equation (3):(9)$$\dot{\varepsilon }exp(\frac{Q}{RT})=A{\left[sinh\left(\alpha \sigma \right)\right]}^{n}\;$$

Figure 7 is the relation curve of lnZ−ln[sinh(ασ)] when the true strain is 0.69.

When ln[sin h(ασ)] was 0, *lnA* was equal to ln Z, and the value was 43.42325.

The constitutive model can be obtained by replacing the above parameters with the hyperbolic sinusoidal constitutive equation:(10)$$\sigma =\frac{1}{\alpha \left(\varepsilon \right)}\cdot ln\{{\left(\frac{Z}{e^{ln\;A\left(\varepsilon \right)}}\right)}^{\frac{1}{n\left(\varepsilon \right)}}+{\left[{\left(\frac{Z}{e^{ln\;A\left(\varepsilon \right)}}\right)}^{\frac{2}{n\left(\varepsilon \right)}}+1\right]}^{\frac{1}{2}}\}\;$$
where Z $=\dot{\varepsilon }\;exp(\frac{Q\left(\varepsilon \right)}{T\cdot 8.314\;J/\left(mol\cdot K\right)})$.

Arrhenius’s constitutive model only thinks about the peak stress but ignores the influence of true strain [25]. To obtain the material constants under different strain variables in the constitutive model, the strain variables are taken from 0.05 to 0.70 with an interval of 0.05. The calculation results of *α, n, Q,* and *lnA* are shown in Table 2.

A polynomial fitting curve of the seventh degree between the strain and the material constants in the constitutive model is shown in Figure 8. Using strain and material constants in the constitutive model *α, n, Q*, and *lnA* in the process of fitting attempt, at least 70% of the calculated points fall on the fitting curve, and the points are evenly distributed on both sides of the curve, guarantee the accuracy of the fitting. Through fitting, it is found that the result of seven times fitting is more accurate. Table 3 displays the parameters in the equation. *A* polynomial of degree seven can express the relationship between *α, n, Q*, *lnA*, and true strain ε, as shown below:(11)$$\alpha \left(\varepsilon \right)=\alpha_{0}+\alpha_{1}\varepsilon +\alpha_{2}\varepsilon^{2}+\alpha_{3}\varepsilon^{3}+\alpha_{4}\varepsilon^{4}+\alpha_{5}\varepsilon^{5}+\alpha_{6}\varepsilon^{6}+\alpha_{7}\varepsilon^{7}\;$$
(12)$$n\left(\varepsilon \right)=n_{0}+n_{1}\varepsilon +n_{2}\varepsilon^{2}+n_{3}\varepsilon^{3}+n_{4}\varepsilon^{4}+n_{5}\varepsilon^{5}+n_{6}\varepsilon^{6}+n_{7}\varepsilon^{7}\;$$
(13)$$Q\left(\varepsilon \right)=Q_{0}+Q_{1}\varepsilon +Q_{2}\varepsilon^{2}+Q_{3}\varepsilon^{3}+Q_{4}\varepsilon^{4}+Q_{5}\varepsilon^{5}+Q_{6}\varepsilon^{6}+Q_{7}\varepsilon^{7}\;$$
(14)$$ln\;A\left(\varepsilon \right)=A_{0}+A_{1}\varepsilon +A_{2}\varepsilon^{2}+A_{3}\varepsilon^{3}+A_{4}\varepsilon^{4}+A_{5}\varepsilon^{5}+A_{6}\varepsilon^{6}+A_{7}\varepsilon^{7}\;$$

The above parameters are substituted into the hyperbolic sinusoidal constitutive equation. The constitutive model of TA1 considering strain shadow is:(15)$$\sigma =\frac{1}{\alpha \left(\varepsilon \right)}\cdot ln\{{\left(\frac{Z}{e^{ln\;A\left(\varepsilon \right)}}\right)}^{\frac{1}{n\left(\varepsilon \right)}}+{\left[{\left(\frac{Z}{e^{ln\;A\left(\varepsilon \right)}}\right)}^{\frac{2}{n\left(\varepsilon \right)}}+1\right]}^{\frac{1}{2}}\}\;$$
where $Z=\dot{\varepsilon }exp(\frac{Q\left(\varepsilon \right)\cdot 1000\;J/mol}{T\cdot 8.314\;J/\left(mol\cdot K\right)})$.

### 3.3. Processing Map of EB Furnace Produced TA1

To reflect the influence of deformation parameters on processing properties, the processing map is drawn to lead the hot deformation process of materials in actual production, where the dynamic materials model (DMM) is generally adopted [26,27]. When the deformation temperature and strain rate remain unchanged, the dynamic relationship between the stress σ and the strain rate of the hot-worked workpiece is as follows:(16)$$\sigma =K{\left(\dot{\varepsilon }\right)}^{m}\;$$

The dynamic material model treats the thermal deformation process as an energy dissipation element. During the plastic deformation of the workpiece, *P* represents the total energy absorbed [28,29]. During plastic deformation, the total energy absorbed can be expressed as [30]:(17)$$P=\sigma \cdot \dot{\varepsilon }=\int_{0}^{\dot{\varepsilon }}\sigma d\dot{\varepsilon }+\int_{0}^{\sigma }\dot{\varepsilon }d\sigma \;$$

The energy absorbed by materials is mainly scattered in two methods: (1) The energy dissipation affected by plastic deformation is expressed as power consumption (*G*), mostly converted to heat; (2) The power dissipation covariance *(J*) is the energy consumed by microstructure evolution [31,32]. The strain rate sensitivity index(m) refers to the proportion of power dissipation co-quantity and power dissipation quantity [33]. m is calculated as follows:(18)$$m=\frac{dJ}{dG}=\frac{d\left(ln\sigma \right)}{d\left(ln\dot{\varepsilon }\right)}=\frac{\Delta lg\;\sigma }{\Delta lg\;\dot{\varepsilon }}\;$$

During the material forming process, the proportional relationship between the energy (*J*) consumed by the evolutionary process of the microstructure and the linear dissipation energy is usually expressed by the power dissipation exponent *η* [34,35]. η is defined as:(19)$$\eta =\frac{J}{J_{max}}=\frac{2m}{m+1}\;$$

The processing instability was established according to the Prasad instability criterion in the dynamic material model, [36], based on Ziegler’s extreme value principle of irreversible thermodynamics in equation (20) [37]: (20)$$\xi =\frac{\partial \;ln(\frac{m}{m+1})}{\partial \;ln\;\dot{\varepsilon }}+m\leq 0\;$$

Along with the established constitutive model, the instability zone and power dissipation index are calculated through Equations (19) and (20). The thermal processing diagram of the dynamic material model is obtained by drawing the principle of superposition of the power dissipation diagram and instability diagram, as shown in Figure 9.

In Figure 9, the gray part of “① and ②” is the plastic instability area. The power dissipation index in this area is less than 0.35, which is not suitable for machining deformation. The dotted line area of “③ and ④” is a suitable processing area. In this area, most of the energy is used for the transformation of the microstructure. The processing map shows that the appropriate hot working areas for industrial pure titanium TA1 are: The range of deformation temperature and strain rates are 700–770 °C and 0.01–0.05 s^−1^, separately; The range of strain rate and deformation temperature is 0.01–0.15 s^−1^ and 840–900 °C, individually. Plastic flow instability region: the range of strain rate and deformation temperature are 3–10 s^−1^ and 860–880 °C, respectively; The range of strain rate and deformation temperature range are 1–10 s^−1^ and 700–840 °C, individually. To effectively reduce the rolling mill load, in combination with the actual industrial production, the processing area ④ is selected for the hot working deformation test of TA1 titanium billet. The preferred 860 °C open rolling annealing organization is shown in the following Figure 10. The structure of finished titanium material is equiaxed α Titanium, the grain size is uniform, and the grain size is about grade 5. The tensile strength of the transverse tension test is 375 MPa, RP_0.2_ is 275 MPa, elongation after fracture is 44%, and the strength and toughness are good.

## 4. Discussion

The VAR process is difficult to eliminate high-density and low-density inclusions during the titanium melting process [6]. In the EB furnace smelting process, the cold hearth can be used to separate the three processes of melting, refining, and crystallization (i.e., melting, refining and solidification). The liquid metal first enters the smelting area for melting and preliminary refining. Then it flows into the refining area for complete refining to eliminate the high-density and low-density inclusions that may be mixed in the raw materials. Finally, the purified melt flows into the crystallizer and condenses into slabs. Therefore, the obtained flat slab phase has a more uniform composition and fewer impurity elements than vacuum consumable smelting [7]. In addition, since the EB furnace works under a vacuum with a pressure less than 0.1 Pa, oxygen and nitrogen impurities formation in the smelting process was suppressed. The single-pass EB process has a good dehydrogenation capacity, and the deformation activation energy and the degree of structural segregation of the EB slab are lower than that of the VAR slab. Li Jun et al. [10] studied that the thermal deformation activation energy of the TA1 sample prepared by VAR is 643.3116 kJ/mol. The thermal deformation activation energy of the TA1 sample prepared by the EB furnace was reduced by about 36.2% compared with the thermal deformation activation energy of the TA1 sample prepared by VAR. By comparing the thermal deformation activation energy, we found that the thermal deformation activation energy of the TA1 sample prepared by EB is lower than that of the sample prepared by VAR. The thermal deformation activation energy of TA1 cast in the EB furnace is 410.655 kJ/mol. Therefore, the TA1 sample prepared by the EB furnace is easier to deform than the TA1 slabs produced by the VAR furnace.

Table 4 compares the processing maps of TA1 slabs produced by the EB furnace and VAR furnace. We found that the machinable range of TA1 slabs produced by VAR is more significant than that produced by EB, whist for the unstable region, the trend is the opposite. The instability area of the processing chart for the slabs produced by EB furnace is relatively large in the region with a large strain rate. Thus, the machinable range of TA1 slab produced by EB is mainly concentrated on a low strain rate, while that of the VAR-produced TA1 slab is relatively concentrated on a high strain rate.

## 5. Conclusions


(1)TA1 prepared by electron beam cold hearth melting in just one pass was very sensitive to strain rate and temperature. The strain rate and temperature of TA1 cast in the EB furnace were very significant. Rising the temperature and lessening the strain rate will reduce the flow stress. In the unsteady deformation stage, the flow stress increased rapidly; The flow stress raised slowly or even fell in the steady-state deformation stage.(2)The constitutive model of TA1 for EB furnace casting considering the effect of strain was Equation (15).(3)Based on the processing map of the TA1 sample produced by the EB furnace, the suitable processing areas were obtained: The range of strain rate was 0.01–0.05 s^−1^ at the deformation temperature range was 700–770 °C; When the deformation temperature range was 840–900 °C, the strain rate range was 0.01–0.15 s^−1^.(4)Compared with VAR produced TA1 sample, EB produced TA1 sample had lower activation energy of thermal deformation and a smaller suitable range for thermal processing.


## Figures and Tables

**Figure 1 materials-16-00369-f001:**
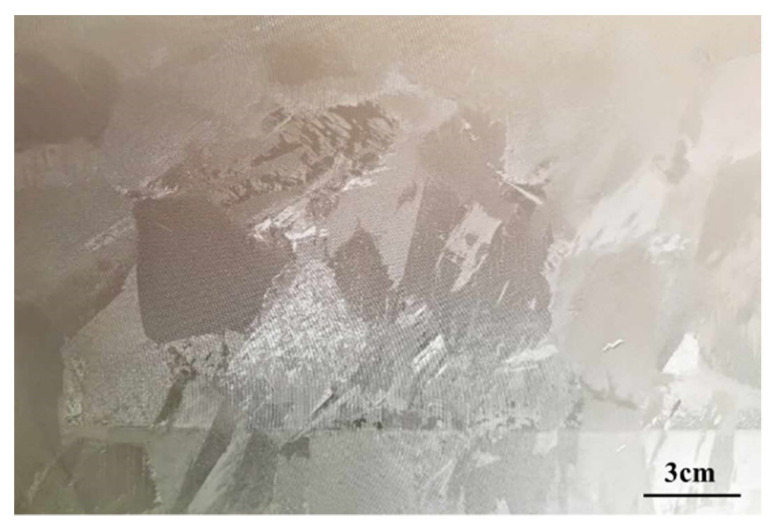
Original structure of slab directly melted and cast in EB furnace.

**Figure 2 materials-16-00369-f002:**
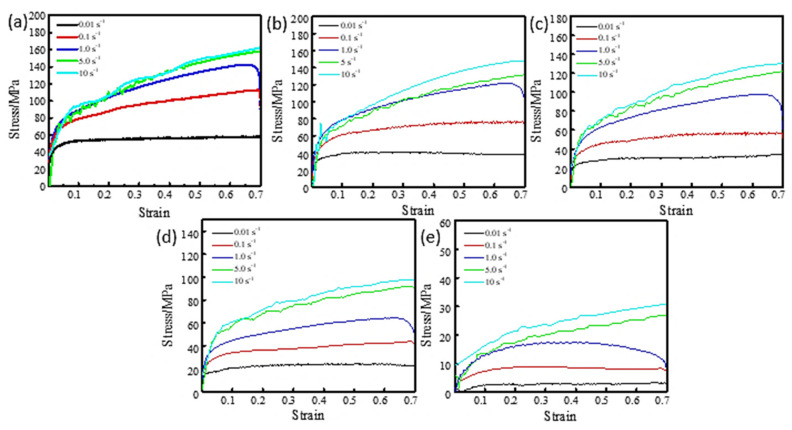
The curves of true stress and true strain at different temperatures: (**a**) 700 °C; (**b**) 750 °C; (**c**) 800 °C; (**d**) 850 °C; (**e**) 900 °C.

**Figure 3 materials-16-00369-f003:**
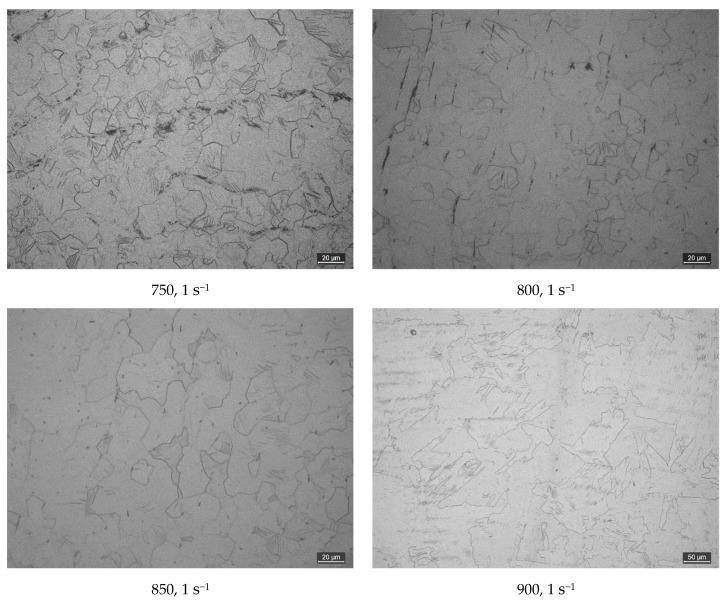
Typical metallographic structure at 750–900 °C and 1 s^−1^ strain rate.

**Figure 4 materials-16-00369-f004:**
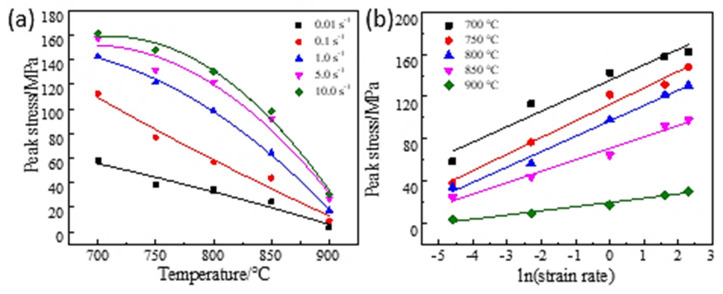
Dependence of peak stress on temperature and strain rate: (**a**) T−σ; (**b**)$\;\ln\;\dot{\varepsilon }-$ σ.

**Figure 5 materials-16-00369-f005:**
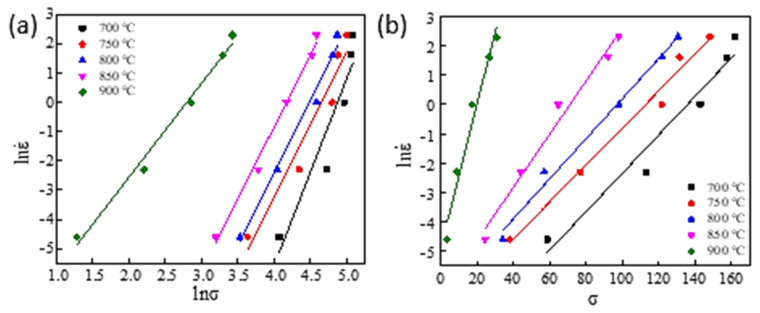
Relationship between (**a**) $\ln\;\dot{\varepsilon }-\ln\sigma$, (**b**) σ$-\ln\;\dot{\varepsilon }$.

**Figure 6 materials-16-00369-f006:**
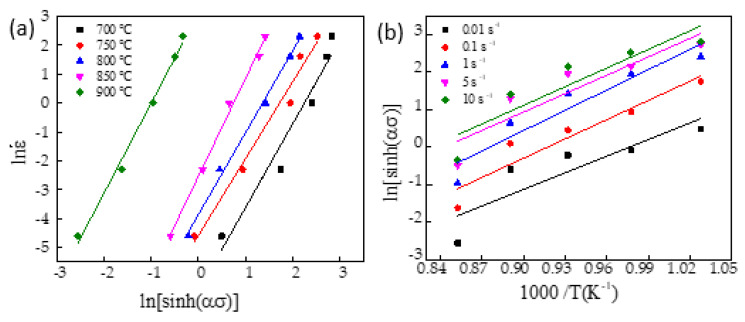
Relationship between (**a**):$\;ln\dot{\varepsilon }-ln\left[sin\;h\left(\alpha \sigma \right)\right]$, (**b**): ln[sin h(ασ)]− 1000/*T*(K^−1^).

**Figure 7 materials-16-00369-f007:**
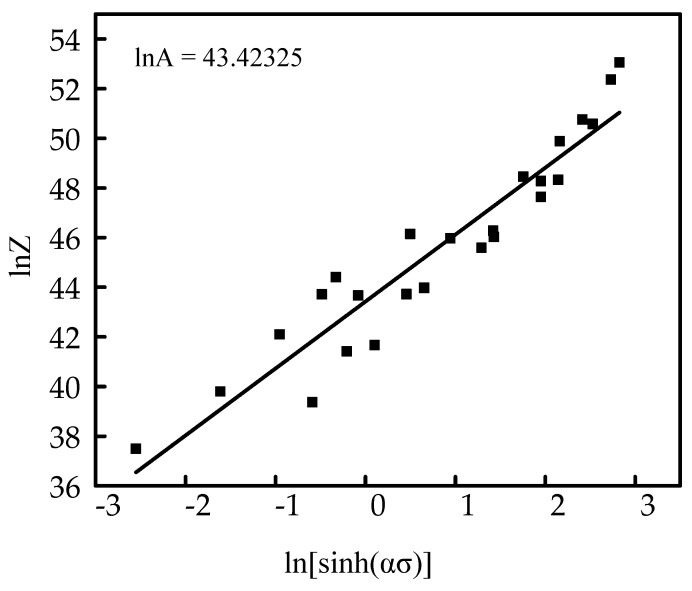
Relationship between ln Z and ln[sin h(ασ)].

**Figure 8 materials-16-00369-f008:**
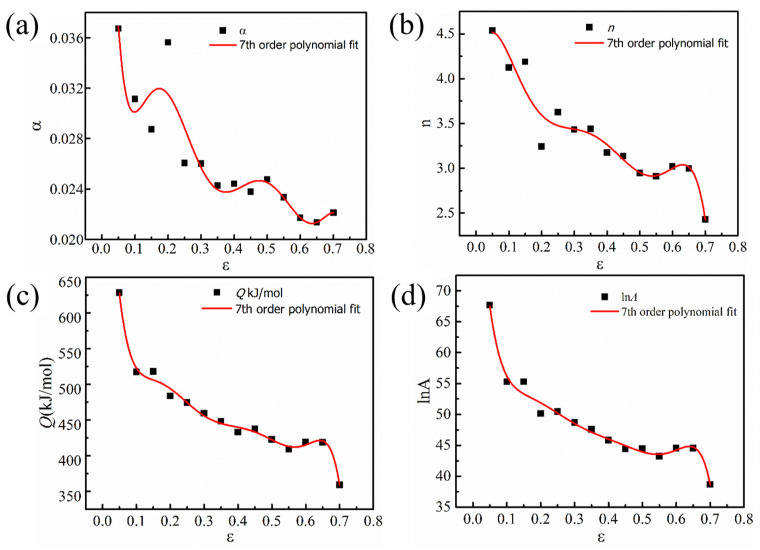
Relationship between material constants and true strain by fitting with seventh-order polynomial: (**a**) *α*; (**b**) *n*; (**c**) *Q*; (**d**) *lnA.*

**Figure 9 materials-16-00369-f009:**
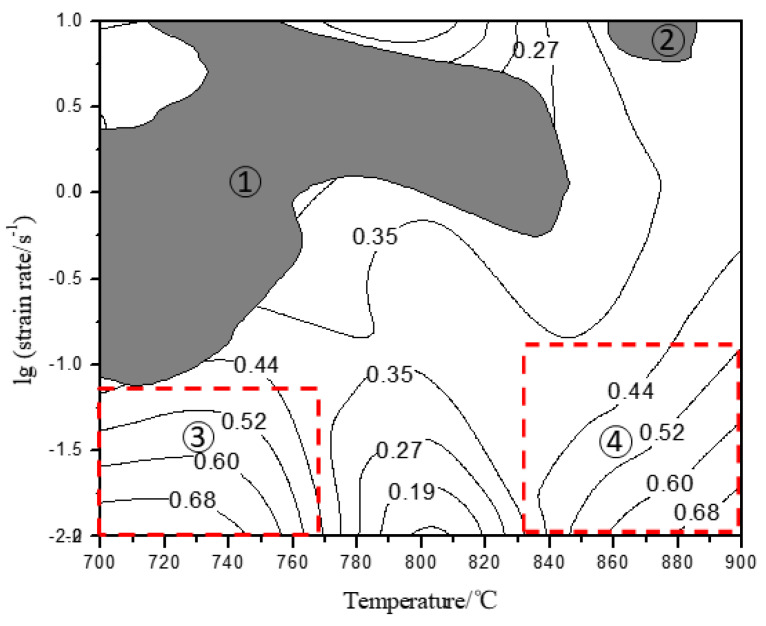
Processing map of TA1. ①, ② are Plastic instability region; ③, ④ are Suitable processing area.

**Figure 10 materials-16-00369-f010:**
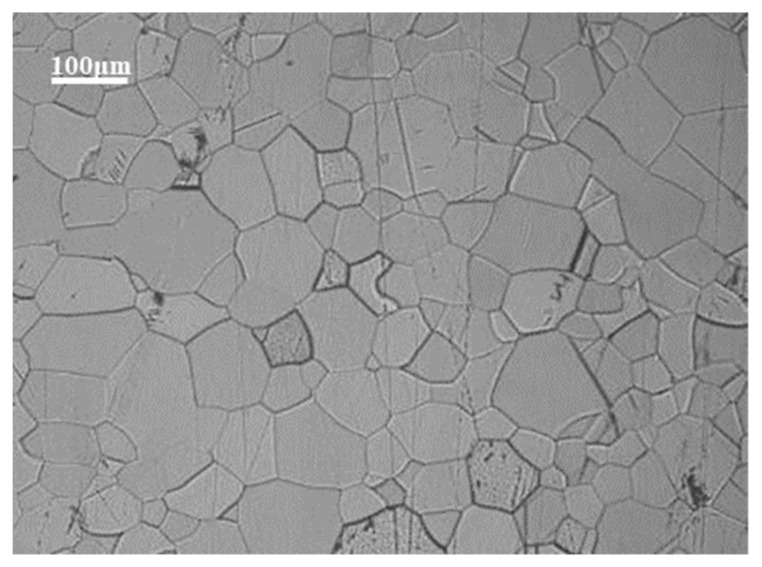
The preferred 860 °C open rolling annealing organization.

**Table 1 materials-16-00369-t001:** Chemical composition of the sample (Wt%).

Fe	O	N	C	H	Ti
≤0.20	≤0.18	≤0.03	≤0.08	≤0.015	Bal

**Table 2 materials-16-00369-t002:** Calculation results of *α, n, Q*, and *lnA*, when the true strain is in the range of 0.05–0.70.

True Strain	*α*	*n*	*Q* (kJ/mol)	*lnA*
0.05	0.0367310	4.537762	628.09658	67.68917
0.1	0.0311358	4.123178	517.24992	55.28302
0.15	0.0287303	4.188236	517.89557	55.30470
0.2	0.0356319	3.242180	483.62878	50.14605
0.25	0.0260507	3.625776	474.42582	50.45797
0.3	0.0260062	3.431418	459.28553	48.67368
0.35	0.0242674	3.438538	448.01972	47.60311
0.4	0.0244092	3.174006	432.84269	45.82066
0.45	0.0237639	3.135992	437.09739	44.38648
0.5	0.0247517	2.947084	422.53110	44.47413
0.55	0.0233285	2.910886	409.14240	43.21425
0.6	0.0216964	3.020520	419.02809	44.54717
0.65	0.0213428	2.996986	418.49972	44.51738
0.7	0.0221200	2.429494	358.97080	38.67918

**Table 3 materials-16-00369-t003:** Data of material constants (*α, n, Q,* and *lnA*).

α	*n*	Q (kJ/mol)	*lnA*
α_0_ = 0.08047	n_0_ = 3.9809	Q_0_ = 1050.80096	A_0_ = 104.97502
α_1_ = −1.57336	n_1_ = 25.52243	Q_1_ = −14,406.71512	A_1_ = −1213.97161
α_2_ = 18.63559	n_2_ = −388.12653	Q_2_ = 156,765.29615	A_2_ = 12,090.3470
α_3_ = −107.21173	n_3_ = 2069.28187	Q_3_ = −890,952.59121	A_3_ = −64,932.77841
α_4_ = 328.63345	n_4_ = −5187.62503	Q_4_ = 2,814,010.0	A_4_ = 198,218.43158
α_5_ = −550.47869	n_5_ = 6070.79641	Q_5_ = 4,982,800.0	A_5_ = −345,319.45288
α_6_ = 475.65222	n_6_ = −2494.63078	Q_6_ = 4,624,390.0	A_6_ = 319,534.03285
α_7_ = −165.87799	n_7_ = −246.27687	Q_7_ = −1,749,330.0	A_7_ = −121,622.70438

**Table 4 materials-16-00369-t004:** Hot deformation of TA1 produced by EB and VAR.

Smelting Method	Suitable Hot Deformation Region
EB	700~760 °C, 0.05~0.01 s^−1^; 840~900 °C, 0.1~0.01 s^−1^;
VAR [10]	750~850 °C, 1~20 s^−1^;
VAR [38]	strain rate < 10 s^−1^, T > 725 °C;
VAR [9]	650~750 °C, 5~10 s^−1^;

## Data Availability

Data presented in this article are available at request from the corresponding author.

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
