# Peer review of "Hot Deformation Behavior of TA1 Prepared by Electron Beam Cold Hearth Melting with a Single Pass"

_materials, 2022, doi:10.3390/ma16010369_

Round 1

Reviewer 1 Report

The manuscript presents the results of basic research focused on hot deformation behavior of TA1. After reviewing the manuscript I would like to address the following comments/suggestions, which should be taken into consideration by the authors:

- In general, the manuscript is prepared very carelessly! It should be carefully revised by the authors before submitting to revision process and publication!

- The manuscript contains huge number of English language errors – even in abstract. It is very difficult to understand due to that. I strongly recommend the revision by an official translator or a native English speaking person! Mandatory.

- The authors should present the microstructure of the investigated TA1 – initial microstructure of the material – before Gleeble testing – and the microstructures of the samples after compression tests. Without it this manuscript cannot be considered as cannot be considered a professional and scientific discussion of hot deformation behavior of the investigated material. It is essential and must be incorporated into the manuscript.

- It is necessary to clarify how the samples were cut out from the ingot. Were they prepared by EDM? What was the direction of sample cutting in relation to the grain orientation in the starting material for testing (TA1 ingot)?

- It is well known that in the case of compression tests friction has a significant effect on the true stress - true strain curve, and the consideration of the friction is strictly required for obtaining correct results. The manuscript does not contain information on that. Were the flow stress curves corrected for friction? 

- Discussing flow stress curves the authors state, that: “When the strain rate is low, the flow stress is low, which is because the sample is affected by dynamic recovery and dynamic recrystallization softening”. How the authors can say that without comparing the microstructures of the samples deformed in compression with the obtained flow stress curves? Such verification of the material flow behavior is essential for proper analysis of the phenomena occurring in the material during hot compression tests.

- The discussion of the obtained results is terrible and short - must be enhanced and corrected. In present state it is very basic. In spite of the basic research presented in this manuscript, there are a lot of issues worth discussing.

- In spite of mentioned above imperfections of this manuscript, a scientific value of the presented results is low.

- Conclusions are very general, very basic. E.g. “The strain rate and temperature of TA1 cast in the EB furnace were very significant” – this is the conclusion? Terrible English and no sense!

Author Response

RESPONSE TO REVIEWERS

Manuscript ID: metals-2009066 “Hot deformation behavior of TA1 prepared by electron beam cold hearth melting with a single pass” by Zhibo Zhang, Weiwei Huang, Weidong Zhao, Xiaoyuan Sun, Haohang Ji, Jin Chen, Lei Gao.

We would like to thank the reviewers for their thoughtful review of the manuscript. They raise important issues and their inputs are very helpful for improving the manuscript. We agree with almost all their comments and we have revised our manuscript accordingly.

We are already crafting a revised version of the paper that states the hypothesis and the implications of our work more clearly than before. Moreover, we are including all reviewers’ suggestions and clarifying the text when needed. We respond below in detail to each of the reviewer’s comments. We hope that the reviewers will find our responses to their comments satisfactory, and we are willing to finish the revised version of the manuscript including any further suggestions that the reviewers may have.

Please, find below the referees’ comments repeated in italics and our responses inserted after each comment.

Looking forward to hearing from you soon.

Sincerely,

Lei Gao

Faculty of Metallurgical and Energy Engineering, Kunming University of Science and Technology, Kunming 650093, China; glkust2013@hotmail.com.

Email of the corresponding author: Science and Technology Innovation Department of Kunming Iron & Steel Co., Ltd. Kunming 650302, China;

The manuscript presents the results of basic research focused on hot deformation behavior of TA1. After reviewing the manuscript. I would like to address the following comments/suggestions, which should be taken into consideration by the authors:

- In general, the manuscript is prepared very carelessly! It should be carefully revised by the authors before submitting to revision process and publication!

Thanks for the reviewer’s valuable comments. We carefully checked and modified the article. The revised places were marked in yellow in the revised manuscript.

- The manuscript contains huge number of English language errors – even in abstract. It is very difficult to understand due to that. I strongly recommend the revision by an official translator or a native English speaking person! Mandatory.

This is very good advice raised by the reviewer. We carefully checked and modified the article. The revised places were marked in yellow in the revised manuscript.

- The authors should present the microstructure of the investigated TA1 – initial microstructure of the material – before Gleeble testing – and the microstructures of the samples after compression tests. Without it this manuscript cannot be considered as cannot be considered a professional and scientific discussion of hot deformation behavior of the investigated material. It is essential and must be incorporated into the manuscript.

Thank the reviewers for their valuable comments. The microstructure of samples before the Gleeble test and the microstructures of the samples after compression tests have been supplemented in Figure 2 and Figure 3 of the manuscript. The supplemented part is marked yellow. Thanks again to the reviewers for their valuable suggestions.

- It is necessary to clarify how the samples were cut out from the ingot. Were they prepared by EDM? What was the direction of sample cutting in relation to the grain orientation in the starting material for testing (TA1 ingot)?

Thanks for the reviewer’s valuable and professional suggestion! According to the comments of the reviewer, we have supplemented the sample cutting method in the material and method section. The samples were cut from the EB furnace directly melted billets produced in the factory with a sawing machine. To ensure the uniformity of the sample taken, remove the end about 10 cm from the end of the blank before cutting. Gleeble sample is processed by wire cutting. The revised places were marked in yellow in the revised manuscript.

- It is well known that in the case of compression tests friction has a significant effect on the true stress - true strain curve, and the consideration of the friction is strictly required for obtaining correct results. The manuscript does not contain information on that. Were the flow stress curves corrected for friction?

This is very good advice raised by the reviewer. We have minimized the effect of friction on the flow stress curve. To reduce the influence of friction, a piece of 0.5mm tantalum foil was placed on both ends of the cylindrical sample during the compression test. Individual data with obvious deviation were removed when processing the flow stress curve. It is supplemented in Article 3.1. The supplementary part has been marked in yellow. Thanks again to the reviewers for their valuable suggestions.

- Discussing flow stress curves the authors state, that: “When the strain rate is low, the flow stress is low, which is because the sample is affected by dynamic recovery and dynamic recrystallization softening”. How the authors can say that without comparing the microstructures of the samples deformed in compression with the obtained flow stress curves? Such verification of the material flow behavior is essential for proper analysis of the phenomena occurring in the material during hot compression tests.

  Thanks for the reviewer’s valuable and professional suggestion! According to the comments of the reviewer, the relevant discussion has been supplemented in the manuscript, and the microstructure of the compression deformation sample has been added in Figure 3.

- The discussion of the obtained results is terrible and short - must be enhanced and corrected. In present state it is very basic. In spite of the basic research presented in this manuscript, there are a lot of issues worth discussing.

We appreciate this suggestion. According to the comments of the reviewer, we have revised the discussion part of the results obtained and marked it in yellow.

- In spite of mentioned above imperfections of this manuscript, a scientific value of the presented results is low.

We appreciate this suggestion. According to the comments of the reviewer, we have revised the discussion part of the results.

- Conclusions are very general, very basic. E.g. “The strain rate and temperature of TA1 cast in the EB furnace were very significant” – this is the conclusion? Terrible English and no sense!

Thanks for the reviewer’s valuable and professional suggestion! We have modified this part to “TA1 prepared by electron beam cold hearth melting in just one pass was very sensitive to strain rate and temperature.”

Reviewer 2 Report

In the paper "Hot deformation behaviour of TA1 prepared by electron beam cold hearth melting with a single pass", the authors have constructed the constitutive model and processing maps for the prediction of the hot deformation behaviour of the titanium alloy. The constructed models show a significant difference in the hot deformation behaviour of the EB and VAR samples. Unfortunately, the paper presents just a technical report of the obtained results without any scientific analysis. The language of the paper is poor. The manuscript is not well written and cannot be accepted in the current state. More specific comments are following:

1.                 It is known, that VAR cannot eliminate the microstructural defects of inclusions, and segregation of composition. These affect significantly cause the hot deformation behaviour. To improve the quality of titanium alloys electron beam cold hearth melting should be improved. However, the authors have shown opposite effect of the EB melting: EB produced TA1 sample had lower activation energy of thermal deformation and a smaller suitable range for thermal processing. This result is very curious and was not explained in the paper.

2.                 The friction between the sample’s edges and the dies such as adiabatic heating during the deformation may significantly influence the true stress – true strain curves [10.1016/j.jallcom.2018.08.010, 10.1179/026708301101510843]. The authors did not consider these effects.

3.                 The obtained stress-strain curves are not informative. It is unclear how did the authors determine the peak stress on the part of them. At the same time the stresses under the 30 MPa (for the chosen diameter of the samples) cannot be measured in the Gleeble 3800 system due to specific installation of the samples in the Hydrowedge II module and the sensitivity of the force measurer.

4.                 The authors have tested as cast samples. The microstructure of the samples tested at different temperatures is significantly differs. The obtained curves cannot be applied for the analysis of the hot deformation behaviour and construction of the constitutive equations and processing maps.

5.                 The authors did not provide any microstructural approvement of the instability or stability deformation in the processing maps regions. The significant microstructural investigations are required to approve the obtained results.

6.                 The paper mostly lacks any discussion and represents just a technical report. The analysis of the obtained results should be significantly wide.

7.                 The initial microstructure such as microstructure after the deformation should be added to the manuscript.

8.                 Exact chemical composition of the investigated alloy should be added to the manuscript.

9.                 The language of the manuscript should be significantly improved.

Author Response

RESPONSE TO REVIEWERS

Manuscript ID: metals-2009066 “Hot deformation behavior of TA1 prepared by electron beam cold hearth melting with a single pass” by Zhibo Zhang, Weiwei Huang, Weidong Zhao, Xiaoyuan Sun, Haohang Ji, Jin Chen, Lei Gao.

We would like to thank the reviewers for their thoughtful review of the manuscript. They raise important issues and their inputs are very helpful for improving the manuscript. We agree with almost all their comments and we have revised our manuscript accordingly.

We are already crafting a revised version of the paper that states the hypothesis and the implications of our work more clearly than before. Moreover, we are including all reviewers’ suggestions and clarifying the text when needed. We respond below in detail to each of the reviewer’s comments. We hope that the reviewers will find our responses to their comments satisfactory, and we are willing to finish the revised version of the manuscript including any further suggestions that the reviewers may have.

Please, find below the referees’ comments repeated in italics and our responses inserted after each comment.

Looking forward to hearing from you soon.

Sincerely,

Lei Gao

Faculty of Metallurgical and Energy Engineering, Kunming University of Science and Technology, Kunming 650093, China; glkust2013@hotmail.com.

Email of the corresponding author: Science and Technology Innovation Department of Kunming Iron & Steel Co., Ltd. Kunming 650302, China;

In the paper "Hot deformation behaviour of TA1 prepared by electron beam cold hearth melting with a single pass", the authors have constructed the constitutive model and processing maps for the prediction of the hot deformation behaviour of the titanium alloy. The constructed models show a significant difference in the hot deformation behaviour of the EB and VAR samples. Unfortunately, the paper presents just a technical report of the obtained results without any scientific analysis. The language of the paper is poor. The manuscript is not well written and cannot be accepted in the current state. More specific comments are following:

  1. It is known, that VAR cannot eliminate the microstructural defects of inclusions, and segregation of composition. These affect significantly cause the hot deformation behaviour. To improve the quality of titanium alloys electron beam cold hearth melting should be improved. However, the authors have shown opposite effect of the EB melting: EB produced TA1 sample had lower activation energy of thermal deformation and a smaller suitable range for thermal processing. This result is very curious and was not explained in the paper.

Thanks for the reviewer’s valuable and professional suggestion! The deformation activation energy Q represents the size of the energy barrier that the atomic transition needs to overcome, and it is an important parameter reflecting the difficulty of alloy deformation. The main factors affecting the deformation activation energy are: 1. Different chemical composition and initial structure (as cast or as forged) of the alloy, and the type, content and structure type of elements affect the deformation degree of the alloy; 2. The selected deformation temperature is different, and the temperature affects the speed of element diffusion in the alloy.

 VAR process is difficult to eliminate high density and low-density inclusions during the titanium melting process. In the EB furnace smelting process, the cold hearth can be used to separate the three processes of melting, refining and crystallization (i.e., melting, refining and solidification). The liquid metal first enters the smelting area for melting and preliminary refining. Then it flows into the refining area for full refining to eliminate the high-density and low-density inclusions that may be mixed in the raw materials. Finally, the purified melt flowing into the crystallizer, and then condense into ingots. Therefore, the obtained flat ingot phase has a more uniform composition and less impurity elements than that of vacuum consumable smelting. In addition, since the EB furnace works under a vacuum condition with a pressure less than 0.1 Pa, the formation of oxygen and nitrogen impurities in the smelting process was suppressed. The single pass EB process has a good dehydrogenation capacity, and the deformation activation energy and the degree of structural segregation of EB ingot are lower than that of the VAR ingot.

  1. The friction between the sample’s edges and the dies such as adiabatic heating during the deformation may significantly influence the true stress – true strain curves [10.1016/j.jallcom.2018.08.010, 10.1179/026708301101510843]. The authors did not consider these effects.

Thanks for the reviewer’s valuable and professional suggestion! We have minimized the effect of friction on the flow stress curve. To reduce the influence of friction, a piece of 0.5 mm tantalum foil was placed on both ends of the cylindrical sample during the compression test. Individual data with obvious deviation were removed when processing the flow stress curve. It is supplemented in article 3.1. The supplementary part has been marked in yellow. Thanks again to the reviewers for their valuable suggestions.

  1. The obtained stress-strain curves are not informative. It is unclear how did the authors determine the peak stress on the part of them. At the same time the stresses under the 30 MPa (for the chosen diameter of the samples) cannot be measured in the Gleeble 3800 system due to specific installation of the samples in the Hydrowedge II module and the sensitivity of the force measurer.

We appreciate this suggestion. The data collected from the metal thermal simulation compression test are sorted out. When the true strain of the sample reaches a certain value, the true stress reaches a peak, which is the peak stress.

  1. The authors have tested as cast samples. The microstructure of the samples tested at different temperatures is significantly differs. The obtained curves cannot be applied for the analysis of the hot deformation behaviour and construction of the constitutive equations and processing maps.

Thanks for the reviewer’s valuable and professional suggestion! By selecting different temperatures and carrying out deformation at different deformation rates, the corresponding stress-strain curves and microstructures are obtained. Based on the obtained stress-strain curve data, the material constants of the studied test samples are derived according to Arrhenius constitutive equation, and the constitutive equation of the test samples is determined. At the same time, based on the dynamic material model, the processing map of the sample is drawn, and finally the deformation temperature and strain rate suitable for processing of the test sample is found. Combined with the industrial production conditions, the industrial test shows that the sample conditions are reliable, and the microstructure and properties of the finished product meet the needs of users. Therefore, the obtained curves can be applied for the analysis of the hot deformation behaviour and construction of the constitutive equations and processing maps.

  1. The authors did not provide any microstructural approvement of the instability or stability deformation in the processing maps regions. The significant microstructural investigations are required to approve the obtained results.

Thanks for the reviewer’s valuable and professional suggestion! According to the comments of the reviewer, the relevant discussion has been supplemented before “When the strain rate is low, the flow stress is low, which is because the sample is affected by dynamic recovery and dynamic recrystallization softening” in the manuscript, and the microstructure of the compression deformation sample has been added in Figure 3.

  1. The paper mostly lacks any discussion and represents just a technical report. The analysis of the obtained results should be significantly wide.

Thanks for the reviewer’s valuable comments. We carefully checked and modified the article. The revised places were marked in yellow in the revised manuscript.

  1. The initial microstructure such as microstructure after the deformation should be added to the manuscript.

Thanks for the reviewer’s valuable and professional suggestion! The microstructure of the compression deformation sample has been added in Figure 3.

  1. Exact chemical composition of the investigated alloy should be added to the manuscript.

Thanks for the reviewer’s valuable comments. The chemical composition has been stated in "2. Materials and Methods". Since the samples were taken from the production samples of the enterprise, the composition is indicated in the mass assurance certificate of the enterprise out of respect for the enterprise and the protection of its technical secrets. Again, we appreciate the comments from the reviewers.

  1. The language of the manuscript should be significantly improved.

Thanks for the reviewer’s valuable comments. According to the comments of the reviewer, we have revised and polished the language of the full text, and the revised part has been marked in yellow.

Reviewer 3 Report

The presented work is variable for engineers. Unfortunately, there are many shortcomings, which must be corrected. The following comments are below:

1.     18 reference is in Chines. It is 60 % of the total reference. The proportion of references in Chine is too large. More papers in English must be added. For example ref. 14 can be replaced by Sun, J.; Chen, Y.; Liu, F.; Yang, E.; Wang, S.; Fu, H.; Qi, Z.; Huang, S.; Yang, J.; Liu, H.; et al. Calibration of Arrhenius Constitutive Equation for B4Cp/6063Al Composites in High Temperatures. Materials (Basel). 2022, 15, 6438, doi:10.3390/ma15186438.

2.     A definition of flow stress should be given.

3.     How was the test of the flow test done? What equipment was used? What geometry of the specimens was used?

4.     A coefficient of determination must be given for estimated equations (11) – (14) from the experiment.

5.     Why was used 7th order polynomial? In paper Xiao, X.; Liu, G.Q.; Hu, B.F.; Zheng, X.; Wang, L.N.; Chen, S.J.; Ullah, A. A comparative study on Arrhenius-type constitutive equations and artificial neural network model to predict high-temperature deformation behaviour in 12Cr3WV steel. Comput. Mater. Sci. 2012, 62, 227–234, doi:10.1016/j.commatsci.2012.05.053 was used 5th order. The coefficient of determination was very high for 5th order polynomials.

 Particular comments are the following:

Line 158 Temperature is in Kelvin degree is indicated in ‘K’ not ‘k’.

Eq. (1) – (6) Variables should be in italic.

Author Response

RESPONSE TO REVIEWERS

Manuscript ID: metals-2009066 “Hot deformation behavior of TA1 prepared by electron beam cold hearth melting with a single pass” by Zhibo Zhang, Weiwei Huang, Weidong Zhao, Xiaoyuan Sun, Haohang Ji, Jin Chen, Lei Gao.

We would like to thank the reviewers for their thoughtful review of the manuscript. They raise important issues and their inputs are very helpful for improving the manuscript. We agree with almost all their comments and we have revised our manuscript accordingly.

We are already crafting a revised version of the paper that states the hypothesis and the implications of our work more clearly than before. Moreover, we are including all reviewers’ suggestions and clarifying the text when needed. We respond below in detail to each of the reviewer’s comments. We hope that the reviewers will find our responses to their comments satisfactory, and we are willing to finish the revised version of the manuscript including any further suggestions that the reviewers may have.

Please, find below the referees’ comments repeated in italics and our responses inserted after each comment.

Looking forward to hearing from you soon.

Sincerely,

Lei Gao

Faculty of Metallurgical and Energy Engineering, Kunming University of Science and Technology, Kunming 650093, China; glkust2013@hotmail.com.

Email of the corresponding author: Science and Technology Innovation Department of Kunming Iron & Steel Co., Ltd. Kunming 650302, China;

The presented work is variable for engineers. Unfortunately, there are many shortcomings, which must be corrected. The following comments are below:

  1. 18 reference is in Chines. It is 60 % of the total reference. The proportion of references in Chine is too large. More papers in English must be added. For example, ref. 14 can be replaced by Sun, J.; Chen, Y.; Liu, F.; Yang, E.; Wang, S.; Fu, H.; Qi, Z.; Huang, S.; Yang, J.; Liu, H.; et al. Calibration of Arrhenius Constitutive Equation for B4Cp/6063Al Composites in High Temperatures. Materials (Basel). 2022, 15, 6438, doi:10.3390/ma15186438.

Thanks for the reviewer’s valuable comments. Reference 14 in the text has been replaced by reference 19, and English references of 14,15,16,17 have been added as references.

  1. A definition of flow stress should be given.

Thanks for the reviewer’s valuable and professional suggestion! The definition of flow stress has been supplemented in section 3.1, and the added content has been marked with yellow.

  1. How was the test of the flow test done? What equipment was used? What geometry of the specimens was used?

Thanks for the reviewer’s professional suggestion! The test material is industrial pure titanium TA1 produced by a factory in Yunnan using an EB furnace for one-time melting and casting. After sampling, it is cut to size by wire Ф 8×12 mm. The isothermal compression test was carried out on samples using a Gleeble − 3800 testing machine.

  1. A coefficient of determination must be given for estimated equations (11) – (14) from the experiment.

Thanks for the reviewer’s valuable comments. Using strain and material constants in the constitutive model α, N, Q, and lnA in the process of fitting attempt, at least 70% of the calculated points fall on the fitting curve, and the points that do not fall on the fitting curve are evenly distributed on both sides of the curve, which means that the fitting is good. This part of the description has been supplemented in the description part of the manuscript figure 8 and has been marked in yellow.

  1. Why was used 7thorder polynomial? In paper Xiao, X.; Liu, G.Q.; Hu, B.F.; Zheng, X.; Wang, L.N.; Chen, S.J.; Ullah, A. A comparative study on Arrhenius-type constitutive equations and artificial neural network model to predict high-temperature deformation behaviour in 12Cr3WV steel. Comput. Mater. Sci. 2012, 62, 227–234, doi:10.1016/j.commatsci.2012.05.053 was used 5th order. The coefficient of determination was very high for 5th order polynomials.

Thanks for the reviewer’s professional suggestion! The experimental model, 0.05 to 0.70, with an interval of 0.05 was used as the strain. Passing material constant α, n, Q, and lnA are calculated, and then corresponding variables and material constants in the constitutive model α, N, Q and lnA tried to fit, and found that the 7th order polynomial fitting is more accurate. Thus, we used 7th order polynomial. The relationship between strain and material constant expressed by several order polynomials varies with different experimental materials and conditions, as well as the analyst's understanding of the degree of fitting. Thanks again to the reviewers for their valuable suggestions.

Particular comments are the following:

Line 158 Temperature is in Kelvin degree is indicated in ‘K’ not ‘k’.

Eq. (1) – (6) Variables should be in italic.

Thanks for the reviewer’s valuable comments. According to the comments of reviewers, "k" has been changed to "K". The variables in equations (1) – (6) are changed to italics.

Reviewer 4 Report

Review on “Hot deformation behaviour of TA1 prepared by electron beam cold hearth melting with a single pass”. Please find the comments on each section to improve the quality of your paper.

1. Introduction

Please highlight the importance of your paper adding a sentence or two on why “impacts of thermal deformation on rheological stress (Line 66)” are so important to study.

2. Materials and methods

a) Table 1: Please add information on how the chemical composition was measured or add the reference to Table.

b) how many specimens were used per one deformation test at a specific temperature and strain rate?

c) how the temperature of the specimen was monitored during the test?

d) what was the heating rate?

e) How the specimen alignment during compression was maintained?

f) How the deformation of the specimen during the test was measured?

3. Results

a) It would be more clear for the reader to have information about the test temperature in the top-right corner

b) would it be possible to unify the stress axis in all graphs from Figure 1 in order to have a clear comparison between the conditions adopted?

c) the description of Figure 2 is too brief. Please add more discussion.

d) In Figure 2b, the Celsius degree is missing.

e) Lines 114-143: please support these two paragraphs with some literature. There is so much discussion without supporting references (for example: Lines 124-126, 130-131, etc).

f) Line 160: please explain how the material constants were determined.

g) It would be beneficial if authors could support their findings summarized in Figure 7 with some papers on the microstructural evolution of TA1 alloy during deformation at such specific conditions. These results should reflect the favourable microstructural changes and mechanisms behind them, especially when the authors defined two suitable hot deformation regions. Please discuss.

4. Discussion

Table 4: The authors have found, that “machinable range of TA1 ingot produced by EB is mainly concentrated on low strain rate, while that of the VAR-produced TA1 ingot is relatively concentrated on high strain rate”. Technically speaking, while using VAR, we can use relatively high strain rates to deform the material and therefore reduce the deformation or forming time significantly. If we compare the deformation strain rates of EB with those of VAR, we can observe almost a hundredfold difference. In this case, why EB method is better than VAR? Please discuss.

Please revise and improve your manuscript according to suggestions putting more emphasis on comments 2a-f, 3g and 4. Please try to polish the language throughout the paper by avoiding informal structures like “constitutive model only thinks about the peak stress but ignores”.

Author Response

RESPONSE TO REVIEWERS

Manuscript ID: metals-2009066 “Hot deformation behavior of TA1 prepared by electron beam cold hearth melting with a single pass” by Zhibo Zhang, Weiwei Huang, Weidong Zhao, Xiaoyuan Sun, Haohang Ji, Jin Chen, Lei Gao.

We would like to thank the reviewers for their thoughtful review of the manuscript. They raise important issues and their inputs are very helpful for improving the manuscript. We agree with almost all their comments and we have revised our manuscript accordingly.

We are already crafting a revised version of the paper that states the hypothesis and the implications of our work more clearly than before. Moreover, we are including all reviewers’ suggestions and clarifying the text when needed. We respond below in detail to each of the reviewer’s comments. We hope that the reviewers will find our responses to their comments satisfactory, and we are willing to finish the revised version of the manuscript including any further suggestions that the reviewers may have.

Please, find below the referees’ comments repeated in italics and our responses inserted after each comment.

Looking forward to hearing from you soon.

Sincerely,

Lei Gao

Faculty of Metallurgical and Energy Engineering, Kunming University of Science and Technology, Kunming 650093, China; glkust2013@hotmail.com.

Email of the corresponding author: Science and Technology Innovation Department of Kunming Iron & Steel Co., Ltd. Kunming 650302, China;

Review on “Hot deformation behaviour of TA1 prepared by electron beam cold hearth melting with a single pass”. Please find the comments on each section to improve the quality of your paper.

  1. Introduction

Please highlight the importance of your paper adding a sentence or two on why “impacts of the rmal deformation on rheological stress (Line 66)” are so important to study.

We appreciate this suggestion. To find out the hot deformation characteristics of EB furnace produced slab and effectively predict and formulate the thermoplastic processing technology of TA1, it is necessary to study the influence of rheological stress during the hot deformation of TA1. We have described this part, and the supplementary part has been marked in yellow.

  1. Materials and methods
  2. a) Table 1: Please add information on how the chemical composition was measured or add the reference to Table.

This is very good advice raised by the reviewer. The chemical composition is indicated in the quality assurance certificate of the enterprise and verified according to the GB/T4698 standard method. The supplementary part has been marked in yellow in the material and method part of the text.

  1. b) how many specimens were used per one deformation test at a specific temperature and strain rate?

Thanks for the reviewer’s valuable comments. One sample shall be used for each specific temperature and strain rate. However, when the deformation of the sample is uneven or the collected data is abnormal, another sample shall be used for confirmation.

  1. c) how the temperature of the specimen was monitored during the test?

Thanks for the reviewer’s valuable comments. During the test, the thermocouple fixed in the middle of the sample is used to measure the temperature, and the data acquisition system of the thermal simulator is used to record the temperature.

  1. d) what was the heating rate?

We appreciate this suggestion. The heating rate is 20 ℃/s.

  1. e) How the specimen alignment during compression was maintained?

Thanks for the reviewer’s valuable and professional suggestion! After the specimen is fixed on the indenter, the position of the specimen shall be corrected to ensure that the specimen is in the center of the indenter and that the specimen is evenly stressed during thermal deformation.

  1. f) How the deformation of the specimen during the test was measured?

This is very good advice raised by the reviewer. The displacement of the sample during compression is recorded by the displacement detector of the thermal simulator, and the feedback signal is sent to the closed loop of the mechanical system to achieve displacement control and recording.

  1. Results
  2. a) It would be more clear for the reader to have information about the test temperature in the top-right corner.

Thanks for the reviewer’s valuable comments. The position of the legend is mainly marked by the position with many blanks according to the trend of the curve in the figure. If you choose to mark all on the upper right corner, the curve of the graph with curves on the upper right corner may be reduced to a certain extent, and it will not be clear enough in a limited location.

  1. b) would it be possible to unify the stress axis in all graphs from Figure 1 to have a clear comparison between the conditions adopted?

We appreciate this suggestion. All figures in Figure 1 can be unified by resetting. Unfortunately, when the specimen is deformed at high temperatures (850 °C, 900 °C), the stress of the specimen is small, and the stress-strain curve will only appear at a very low position in the lower part of the figure. The author thinks that this is not conducive to clearly showing the stress change process at this temperature. Again, we appreciate the comments from the reviewers.

  1. c) the description of Figure 2 is too brief. Please add more discussion.

Thanks for the reviewer’s valuable comments. For the description of Figure 2, we have made a supplement, and the supplementary part has been marked in yellow (Figure 4).

  1. d) In Figure 2b, the Celsius degree is missing.

Thanks for the reviewer’s valuable comments. We have added Celsius in Figure 4b.

  1. e) Lines 114-143: please support these two paragraphs with some literature. There is so much discussion without supporting references (for example: Lines 124-126, 130-131, etc).

Thanks for the reviewer’s valuable and professional suggestion! According to the comments of reviewers, we have supplemented relevant references ([15,16,17]) at Lines 124-126, 130-131 and 136-137. The compensation part has been marked yellow.

  1. f) Line 160: please explain how the material constants were determined.

Thanks for the reviewer’s valuable comments. By taking the natural logarithms on both sides of equations (1), (2) and (3) respectively, we can get equation (4), equation (5), and equation (6). From the equation, we can see that n1 is (σ−) The slope of the relationship, β That is (σ−) The slope of the relationship. By drawing the relationship diagram (Fig. 3a) with  as the abscissa and  as the ordinate, the average slope is n1; Draw a diagram (Fig. 3b) with σ as the abscissa and  as the ordinate, and the average slope is β. So, α can be calculated.

  1. g) It would be beneficial if authors could support their findings summarized in Figure 7 with some papers on the microstructural evolution of TA1 alloy during deformation at such specific conditions. These results should reflect the favorable microstructural changes and mechanisms behind them, especially when the authors defined two suitable hot deformation regions. Please discuss.

Thanks for the reviewer’s valuable comments. To effectively reduce the load of the rolling mill, in combination with the actual industrial production, the processing area â‘£ is selected for the hot working deformation test of TA1 titanium slab. The preferred 860 ℃ open rolling annealing organization is shown in the following figure. The structure of finished titanium material is equiaxed α Titanium, the grain size is uniform, and the grain size is about grade 5. The tensile strength of the transverse tension test is 375 MPa, RP0.2 is 275 MPa, elongation after fracture is 44%, and the strength and toughness are good. The revised places were marked in yellow in the revised manuscript. Again, we appreciate the comments from the reviewers.

  1. Discussion

Table 4: The authors have found, that the “machinable range of TA1 slab produced by EB is mainly concentrated on low strain rate, while that of the VAR-produced TA1 slab is relatively concentrated on high strain rate”. Technically speaking, while using VAR, we can use relatively high strain rates to deform the material and therefore reduce the deformation or forming time significantly. If we compare the deformation strain rates of EB with those of VAR, we can observe almost a hundredfold difference. In this case, why EB method is better than VAR? Please discuss.

Thanks for the reviewer’s valuable and professional suggestion! The advantage of using EB furnace slabs as raw materials for titanium processing is not the speed of strain rate compared with VAR furnace slabs, but that EB furnace slabs greatly reduce the cost of titanium processing raw materials. According to the production calculation, the VAR furnace needs to melt and cast slabs at least twice, and then forge them into titanium slab raw materials. There are many processing procedures, and long processes, and the processing cost is about 25000 yuan(RMB)/ton. EB furnace uses one time melting and casting to form directly, and the processing cost is about 17000 yuan(RMB)/ton. In addition, the processing procedures and processing time are reduced, which can greatly shorten the supply cycle. This is very obvious for the advantages of actual production enterprises.

The original intention of this paper is to provide a reference for the subsequent hot deformation processing of titanium slabs smelted in a more economical EB furnace, not to prove that EB slabs are better than VAR slabs. Again, we appreciate the comments from the reviewers.

Please revise and improve your manuscript according to suggestions putting more emphasis on comments 2a-f, 3g, and 4. Please try to polish the language throughout the paper by avoiding informal structures like “constitutive model only thinks about the peak stress but ignores”.

Thanks for the reviewer’s valuable comments. We carefully checked and modified the article. The revised places were marked in yellow in the revised manuscript.

---------------------------------------------------------------------------------------------------------------

Special thanks to you for your valuable comments. Please point out any deficiencies in our work or the revised manuscript, and we would like to revise the manuscript according to your comments until it meets the publishing requirements.

Round 2

Reviewer 1 Report

The authors made necessary corrections according to the suggestions, which made the quality of this manuscript higher. There are some imperfections, that still should be corrected, like the quality of Figure 1.

Author Response

RESPONSE TO REVIEWERS

Manuscript ID: metals-2009066 “Hot deformation behavior of TA1 prepared by electron beam cold hearth melting with a single pass” by Zhibo Zhang, Weiwei Huang, Weidong Zhao, Xiaoyuan Sun, Haohang Ji, Jin Chen, Lei Gao.

We would like to thank the reviewers for their thoughtful review of the manuscript. They raise important issues and their inputs are very helpful for improving the manuscript. We agree with almost all their comments and we have revised our manuscript accordingly.

We are already crafting a revised version of the paper that states the hypothesis and the implications of our work more clearly than before. Moreover, we are including all reviewers’ suggestions and clarifying the text when needed. We respond below in detail to each of the reviewer’s comments. We hope that the reviewers will find our responses to their comments satisfactory, and we are willing to finish the revised version of the manuscript including any further suggestions that the reviewers may have.

Please, find below the referees’ comments repeated in italics and our responses inserted after each comment.

Looking forward to hearing from you soon.

Sincerely,

Lei Gao

Faculty of Metallurgical and Energy Engineering, Kunming University of Science and Technology, Kunming 650093, China; glkust2013@hotmail.com.

Email of the corresponding author: Science and Technology Innovation Department of Kunming Iron & Steel Co., Ltd. Kunming 650302, China;

There are some imperfections, that still should be corrected, like the quality of Figure 1.

Thanks for the reviewer’s valuable comments. We carefully checked and modified the article where the qualities of the figures were included. The revised places were marked in yellow in the revised manuscript.

Reviewer 2 Report

The authors have answered part of the previous comments. However, some of the parts in the manuscript are still questionable:

1.                 It is unclear, how the authors have determined the peak stress for the curves for low temperatures and high strain rates. The stress does not achieve a maximum on the most of them. Obtained constitutive equations are not correct. Only true peak stresses should be used for the model construction.

2.                 The quality of the initial microstructure is too low.

3.                 “A piece of 0.5 mm tantalum foil” cannot decrease the influence of the friction. It plays a role as thermal insulator for the uniform temperature distribution in the sample during hot deformation. The friction correction of the curves by measuring “barrel” of the samples after the compression is required. Adiabatic heating during the deformation at high strain rates is also ignored. Please, correct the curves for these effects (10.1016/j.jallcom.2018.08.010, 10.1179/026708301101510843).

Author Response

RESPONSE TO REVIEWERS

Manuscript ID: metals-2009066 “Hot deformation behavior of TA1 prepared by electron beam cold hearth melting with a single pass” by Zhibo Zhang, Weiwei Huang, Weidong Zhao, Xiaoyuan Sun, Haohang Ji, Jin Chen, Lei Gao.

We would like to thank the reviewers for their thoughtful review of the manuscript. They raise important issues and their inputs are very helpful for improving the manuscript. We agree with almost all their comments and we have revised our manuscript accordingly.

We are already crafting a revised version of the paper that states the hypothesis and the implications of our work more clearly than before. Moreover, we are including all reviewers’ suggestions and clarifying the text when needed. We respond below in detail to each of the reviewer’s comments. We hope that the reviewers will find our responses to their comments satisfactory, and we are willing to finish the revised version of the manuscript including any further suggestions that the reviewers may have.

Please, find below the referees’ comments repeated in italics and our responses inserted after each comment.

Looking forward to hearing from you soon.

Sincerely,

Lei Gao

Faculty of Metallurgical and Energy Engineering, Kunming University of Science and Technology, Kunming 650093, China; glkust2013@hotmail.com.

Email of the corresponding author: Science and Technology Innovation Department of Kunming Iron & Steel Co., Ltd. Kunming 650302, China;

1.It is unclear, how the authors have determined the peak stress for the curves for low temperatures and high strain rates. The stress does not achieve a maximum on the most of them. Obtained constitutive equations are not correct. Only true peak stresses should be used for the model construction.

Thanks for the reviewer’s valuable comments. The data collected are indeed true stresses under true strain conditions. After sorting out the collected data, when the true strain of the sample reaches a certain value, the true stress reaches a peak value, which is taken as the peak stress.

2.The quality of the initial microstructure is too low.

Thanks for the reviewer’s valuable comments. We carefully checked and modified the article where the qualities of the figures were included. The revised places were marked in yellow in the revised manuscript.

  1. “A piece of 0.5 mm tantalum foil” cannot decrease the influence of the friction. It plays a role as thermal insulator for the uniform temperature distribution in the sample during hot deformation. The friction correction of the curves by measuring “barrel” of the samples after the compression is required. Adiabatic heating during the deformation at high strain rates is also ignored. Please, correct the curves for these effects (10.1016/j.jallcom.2018.08.010, 10.1179/026708301101510843).

 We have carefully studied the recommended papers (10.1016/j.jallcom.2018.08.010, 10.1179/026708301101510843). Some comments in these articles are listed:

[13] The friction effect in the process of thermal deformation is reduced by placing graphite sheets at both ends of the sample (High quality graphite sheet with thickness of 0.05 mm was utilized between anvils and specimen surface to reduce the friction effect during hot deformation.)

[14] The most important variables are the friction force and the geometry of the sample. (Strain and strain rate are of intermediate importance with volume and temperature the least important.)

In the process of carrying out this experiment, in order to reduce the influence of friction, both ends of the indenter of the testing machine were coated with graphite emulsion according to conventional operations during the compression experiment. In addition, a 0.5mm tantalum metal foil is used to reduce the impact of friction on data. When the deformation of the sample is uneven, that is, the obtained sample has irregular "drum shape" or even non-drum shape, the collected data can no longer truly characterize the stress-strain relationship, and the corresponding process can be re-tested. When processing the flow stress curve, the offset data were removed and corrected by referring to [13][14].

[13]Zhipeng Wan, Lianxi Hu , Yu Sun,et al. Hot deformation behavior and processing workability of a Ni-based alloy. Journal of Alloys and Compounds,2018,769:367-375.

[14]R. W. Evans , P. J. Scharning,Axisymmetric compression test and hot orking properties of alloys, Materials Science and Technology , 2001 , 17: 995-1004.

Reviewer 3 Report

The authors corrected the manuscript according to the recommendation. I accept to publish.

Author Response

Thank you

Reviewer 4 Report

The manuscript has been improved according to reviewer suggestions thus it could be considered for publication. However, some issues should be considered before publication.

1. The technical detail about mechanical testing should be included in the text. These include how the temperature was monitored, what was the heating rate etc.

2. If the authors decide to test one specimen per condition, then it should be explained why it is enough for their study. Furthermore, the authors claimed that "when the deformation of the sample is uneven or the collected data is abnormal, another sample shall be used for confirmation."  - please explain what it means if the deformation is uneven or data is abnormal? The justification should be included in the manuscript text as well.

3. Figure 1 is blurry. It looks like the photo from the photograph. It should be replaced by the proper one.

4. Figure 3. Would it be possible to improve the brightness/contrast in order to have something similar to Figure 10? In the current version of the Fig.3, the grains are hardly observed. Also, it would be beneficial to calculate the average grain size of each temperature. Then the recrystallization/phase transformation/grain growth would be easier to observe.

5. Please include the response on discussion section and support it with proper references. The authors' response is important to make the paper's aim clear.

6. Almost half of the references are in the Chinese language thus it is hard to verify or refer to them for non-chinese speakers. Since Metals is an international journal, please try to refer to or add more papers that could support authors claims.

Author Response

RESPONSE TO REVIEWERS

Manuscript ID: metals-2009066 “Hot deformation behavior of TA1 prepared by electron beam cold hearth melting with a single pass” by Zhibo Zhang, Weiwei Huang, Weidong Zhao, Xiaoyuan Sun, Haohang Ji, Jin Chen, Lei Gao.

We would like to thank the reviewers for their thoughtful review of the manuscript. They raise important issues and their inputs are very helpful for improving the manuscript. We agree with almost all their comments and we have revised our manuscript accordingly.

We are already crafting a revised version of the paper that states the hypothesis and the implications of our work more clearly than before. Moreover, we are including all reviewers’ suggestions and clarifying the text when needed. We respond below in detail to each of the reviewer’s comments. We hope that the reviewers will find our responses to their comments satisfactory, and we are willing to finish the revised version of the manuscript including any further suggestions that the reviewers may have.

Please, find below the referees’ comments repeated in italics and our responses inserted after each comment.

Looking forward to hearing from you soon.

Sincerely,

Lei Gao

Faculty of Metallurgical and Energy Engineering, Kunming University of Science and Technology, Kunming 650093, China; glkust2013@hotmail.com.

Email of the corresponding author: Science and Technology Innovation Department of Kunming Iron & Steel Co., Ltd. Kunming 650302, China;

  1. The technical detail about mechanical testing should be included in the text. These include how the temperature was monitored, what was the heating rate etc.

We appreciate this suggestion. The temperature is measured by a thermocouple fixed in the middle of the sample. The temperature was recorded by the data acquisition system of the thermal simulator. The heating rate is 20℃/s.

  1. If the authors decide to test one specimen per condition, then it should be explained why it is enough for their study. Furthermore, the authors claimed that "when the deformation of the sample is uneven or the collected data is abnormal, another sample shall be used for confirmation." - please explain what it means if the deformation is uneven or data is abnormal? The justification should be included in the manuscript text as well.

The technology of hot compression deformation test using Gleeble testing machine is relatively mature. Therefore, using one sample for each specific temperature and strain rate can meet the research requirements. But it also depends on the ability of the tester to control the machine. When the sample deformation is not uniform, that is, the obtained sample has irregular "drum shape" or even non-" drum shape ", the collected data will also appear obvious anomalies. At this point, another sample will be made to correct the corresponding process.

  1. Figure 1 is blurry. It looks like the photo from the photograph. It should be replaced by the proper one.

We appreciate this suggestion. We updated the quality of the figure. However, because the original grain size of the EB furnace melting cast titanium billet is centimeter-level, the micro-morphology with a large number of grains cannot be obtained by conventional metallography microscope. Considering that pure titanium is α-Ti at room temperature and its microstructure is a simple equiaxed grain, the macroscopic microstructure was obtained by stereopicroscope. The microstructure is different from that obtained by conventional metallographic microscopes.

  1. Figure 3. Would it be possible to improve the brightness/contrast in order to have something similar to Figure 10? In the current version of the Fig.3, the grains are hardly observed. Also, it would be beneficial to calculate the average grain size of each temperature. Then the recrystallization/phase transformation/grain growth would be easier to observe.

We appreciate this suggestion. Figure 10 shows the microstructure after hot rolling annealing. After annealing, the grain is fully recovered, so the grain boundary is clear and easy to be corroded. FIG. 3 shows the microstructure obtained after direct deformation and cooling under the conditions of set temperature and strain rate. In some deformation conditions, the grains are in recrystallization state, and the recovery is not sufficient, and the grains are difficult to corrode. The author tried to adjust the contrast and brightness of the photos so that the grain can be more clearly presented to the reader.

  1. Please include the response on discussion section and support it with proper references. The authors' response is important to make the paper's aim clear.

We appreciate this suggestion. As suggested by the reviewer, we add these information in the discussion section with updated references.

  1. Almost half of the references are in the Chinese language thus it is hard to verify or refer to them for non-chinese speakers. Since Metals is an international journal, please try to refer to or add more papers that could support authors claims.

We appreciate this suggestion. According to the opinions of the reviewers, more English papers have been referred to, see the reference section of the article for details. However, this paper is highly connected with the progress of EB experiments in China. We also want to show this progress, which supported by some articles published on Chinese Academic journal. We still want to keep some of the key references which covers important information in their English Abstract.  

---------------------------------------------------------------------------------------------------------------

Special thanks to you for your valuable comments. Please point out any deficiencies in our work or the revised manuscript, and we would like to revise the manuscript according to your comments until it meets the publishing requirements.